# Anatomy of leaf apical hydathodes in four monocotyledon plants of economic and academic relevance

**Alain Jauneau[1]⊗*, Aude Cerutti[2]⊗, Marie-Christine Auriac[1,2], Laurent D. Noël[ID][2]***

**1** Fédération de Recherche 3450, Université de Toulouse, CNRS, Université Paul Sabatier, Castanet-Tolosan, France, **2** LIPM, Université de Toulouse, INRAE, CNRS, Université Paul Sabatier, Castanet-Tolosan, France

⊗ These authors contributed equally to this work.
* jauneau@lrsv.ups-tlse.fr (AJ); laurent.noel@inrae.fr (LN)

**Data Availability Statement:** All relevant data are within the manuscript and its Supporting Information files.

## Abstract

Hydathode is a plant organ responsible for guttation in vascular plants, i.e. the release of droplets at leaf margin or surface. Because this organ connects the plant vasculature to the external environment, it is also a known entry site for several vascular pathogens. In this study, we present a detailed microscopic examination of leaf apical hydathodes in monocots for three crops (maize, rice and sugarcane) and the model plant *Brachypodium distachyon*. Our study highlights both similarities and specificities of those epithemal hydathodes. These observations will serve as a foundation for future studies on the physiology and the immunity of hydathodes in monocots.

## Introduction

Guttation is the physiological release of fluids in the aerial parts of the plants such as leaves, sepals and petals. This phenomenon can be the result of local active water release by specialized cells or organs in so-called active hydathodes such as trichomes or glands. In contrast, passive hydathodes (also named epithemal hydathodes) are organs in which guttation is mostly driven by the root pressure [for review, see 1]. Guttation at passive hydathodes is usually observed in conditions where stomata are closed and humidity is high. Such guttation is supposed to play an important role in plant physiology to promote water movement *in planta* in specific conditions [2, 3], to detoxify plant tissues by exporting excessive salts or molecules [4, 5] and to specifically capture some solutes from xylem sap before guttation [6]. These passive hydathodes thus appear as an interface between the plant vasculature and the outside.

Passive hydathodes can be found at the leaf tip (apical hydathodes), on the leaf blade (laminar hydathodes) and at the leaf margin (marginal hydathodes) depending on the plant family [for review, see 1]. Despite this diversity, passive hydathodes share a conserved anatomy: i) epidermal water pores, resembling stomata at the surface, ii) a parenchyma called the epithem, composed of small loosely connected cells and many intercellular spaces and iii) a hypertrophied and branched xylem system irrigating the epithem [7, 8]. In some plants, the epithem

**Funding:** AJ and LDN are supported by the NEPHRON project (ANR-18-CE20-0020-01). This work was supported by a PhD grant from the French Ministry of National Education and Research to AC. LIPM is part of the TULIP LabEx (ANR-10-LABX-41; ANR-11-IDEX-0002-02).

**Competing interests:** The authors have declared that no competing interests exist.

may be physically separated from the mesophyll by a bundle sheath or a compact layer of cells called tanniferous bundle [7].

Hydathodes are also relevant to plant health because they represent natural entry points for several vascular bacterial pathogens in both monocot and dicot plants. Hydathode infection is visible by chlorotic and necrotic symptoms starting at leaf tips or leaf margins leading to systemic infections as observed in black rot of Brassicaceae caused by *Xanthomonas campestris* pv. *campestris* [9], in bacterial blight of aroids caused by *Xanthomonas axonopodis* pv. *dieffenbachiae* [10, 11], in bacterial canker of tomato caused by *Clavibacter michiganensis* subsp. *michiganensis* [12] and in bacterial leaf blight of rice caused by *X. oryzae* pv. *oryzae* (*Xoo*) [13–16]. Certain pathogens are thus adapted to colonize the hydathode niche and access plant vasculature.

Though hydathodes were first described over a century ago, their anatomy is still poorly described. Most published studies use single microscopic techniques and provide descriptions of either surface or inner organizations so that a global overview of the organ is difficult to capture. Because most of the anatomic studies were performed before the 80s, literature search engines such as Pubmed will not lead you to such publications. Anatomy of arabidopsis hydathodes has only been recently reported [9]. Only scarce descriptions are available for monocot hydathodes, and none in the model plant *Brachypodium distachyon*. In rice (*Oryza sativa*) hydathodes, the large vessel elements are not surrounded by a bundle sheath but included in a lacunar mesophyll facing water pores [17, 18]. In barley (*Hordeum vulgare*), a single hydathode is also found at the leaf tip and water pores are reported to be very close to vascular elements [19]. In wheat (*Triticum aestivum*), an ultrastructural study showed that intercellular space directly connects vessel elements with water pores [17]. Determining or refining the anatomy of hydathodes in those or other monocots is a thus a prerequisite to study the physiology and the immunity of those organs.

In this study, we report on the anatomy of hydathodes in four species of monocots, such as rice, sugarcane, maize and the model plant *Brachypodium distachyon* using a combination of optical and electron microscopy on fresh or fixed tissues. Our study highlights both similarities and specificities of those epithemal hydathodes and provides a comprehensive overview of their anatomy.

## Results

### SEM observations of leaf tips in four monocot plants reveals the presence of water pores anatomically distinct from stomata

Guttation was observed at leaf tips in maize, rice, *Brachypodium* and sugarcane indicating the presence of apical hydathodes (Figs 1A, 2A, 3A and 4A). Though guttation at leaf margins can also be observed in sugarcane (Fig 4A), we did not study marginal hydathodes in this manuscript. In order to characterize apical hydathodes in these four plants, we first observed leaf tips by scanning electron microscopy (SEM). Leaves of rice (Fig 2B), *Brachypodium* (Fig 3B and 3I) and sugarcane (Fig 4E) present elongated and thin tips compared to maize (Fig 1A–1B). All the leaf tips form a more or less pronounced gutter and are decorated by trichomes (Figs 1B, 2B, 3B, 3C, 3H, 3I and 4B). Rice leaf tip and blade are also covered on both faces by round-shaped spicules (Fig 2B–2F). At a smaller scale, numerous grooves and depressions associated to epidermal cell junctions are observed (Figs 1C, 2C, 3B–3E and 4B and 4C).

On both sides of the leaf tips, numerous pores made of pairs of guard cells can be observed though sometimes with difficulty when located deep in the groove on the adaxial face of the leaf (Figs 1B–1D, 2C, 2D, 3B-3E, 4B and 4C). In maize, rice and *Brachypodium*, such pores were only observed within 500 μm from the tip where guttation happens and are likely water

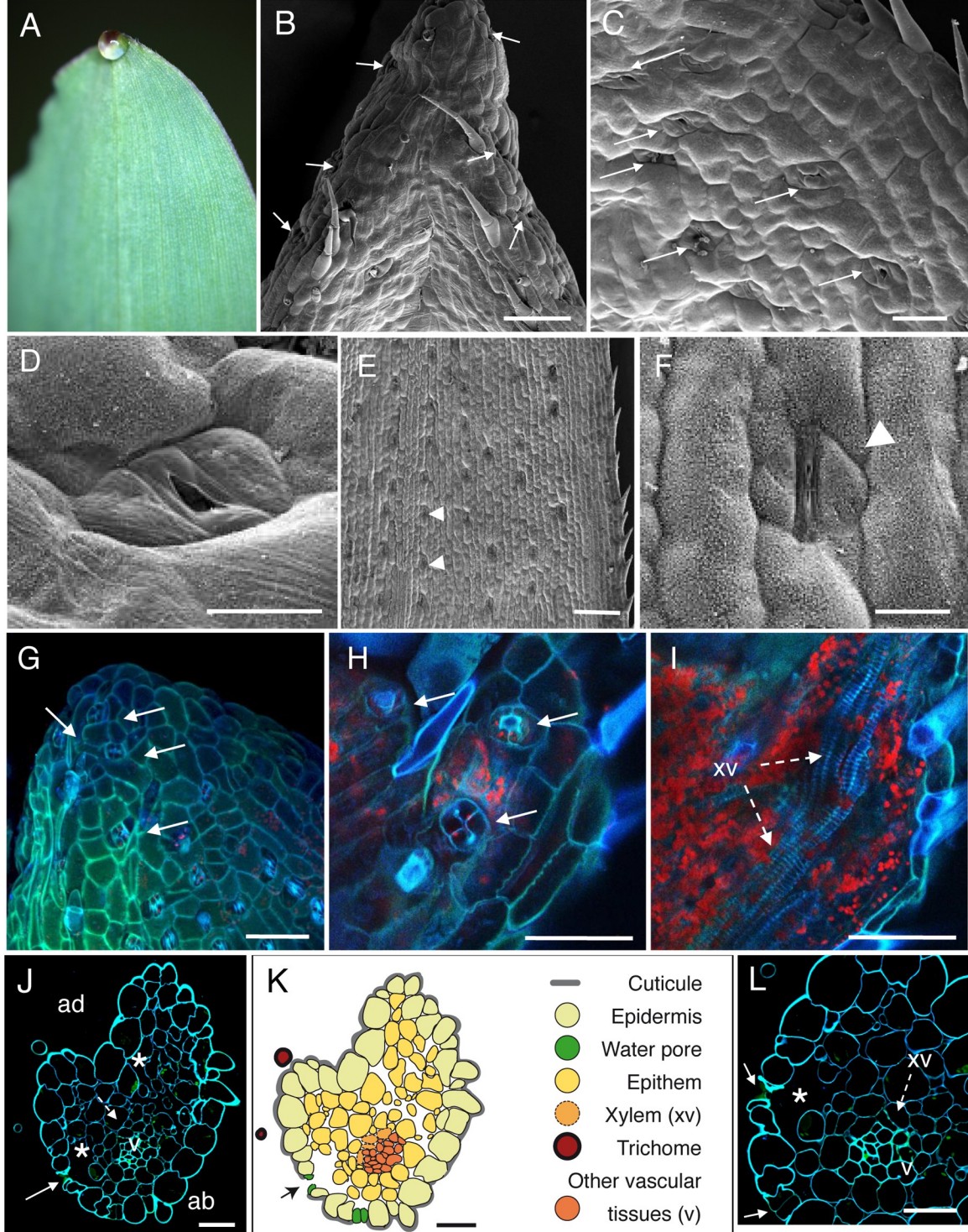

**Fig 1. Anatomic description of maize apical hydathodes by confocal and Scanning Electron Microscopy (SEM).** (A) A guttation droplet at the leaf tip. (B-D) The adaxial face of leaf tip was imaged by SEM. Water pores are observed in the gutter. (E-F) Observations of stomata were performed on distal parts of leaves relative to panels B-D. Panel F is a closeup image of panel E. (G-I) Confocal images of fresh adaxial face of the leaf tip. (G) The image is a maximal projection of 50 confocal planes in z dimension (1-μm steps). (H-I) Observations in z axis of the same sample at the epidermal level (H) and below the epidermal layer (I). Each overlay image corresponds to the maximal projection of 25–30 confocal planes acquired in z dimension. (J, L) Transversal sections (1-μm thickness) of fixed tissue at 80–100 μm from the tip were observed by confocal microscopy. White arrows, arrowheads, dashed arrows and asterisks indicate water

pores, stomata, xylem vessels (xv) and large chambers and intercellular spaces, respectively. v, small veins; ad, adaxial face; ab, abaxial face. (K) Schematic drawing of the hydathode cross section observed in J. Scale bars: B, E, G-I: 100 μm; F: 20 μm; C: 40 μm; D, J-L: 30μm.

pores. In sugarcane, the spike epidermis does not present water pores until ~800 μm from the tip (Fig 4C). Water pore features were also determined in parallel by the observation of fresh leaf tips mounted in water using confocal microscopy taking advantage of tissue autofluorescence (Figs 1G, 1H, 2G, 3H-3J and 4E-4G). We observed that the following criteria could discriminate water pores from stomata (Figs 1E, 1F, 2E, 2F, 3F, 3G and 4D): their location at the

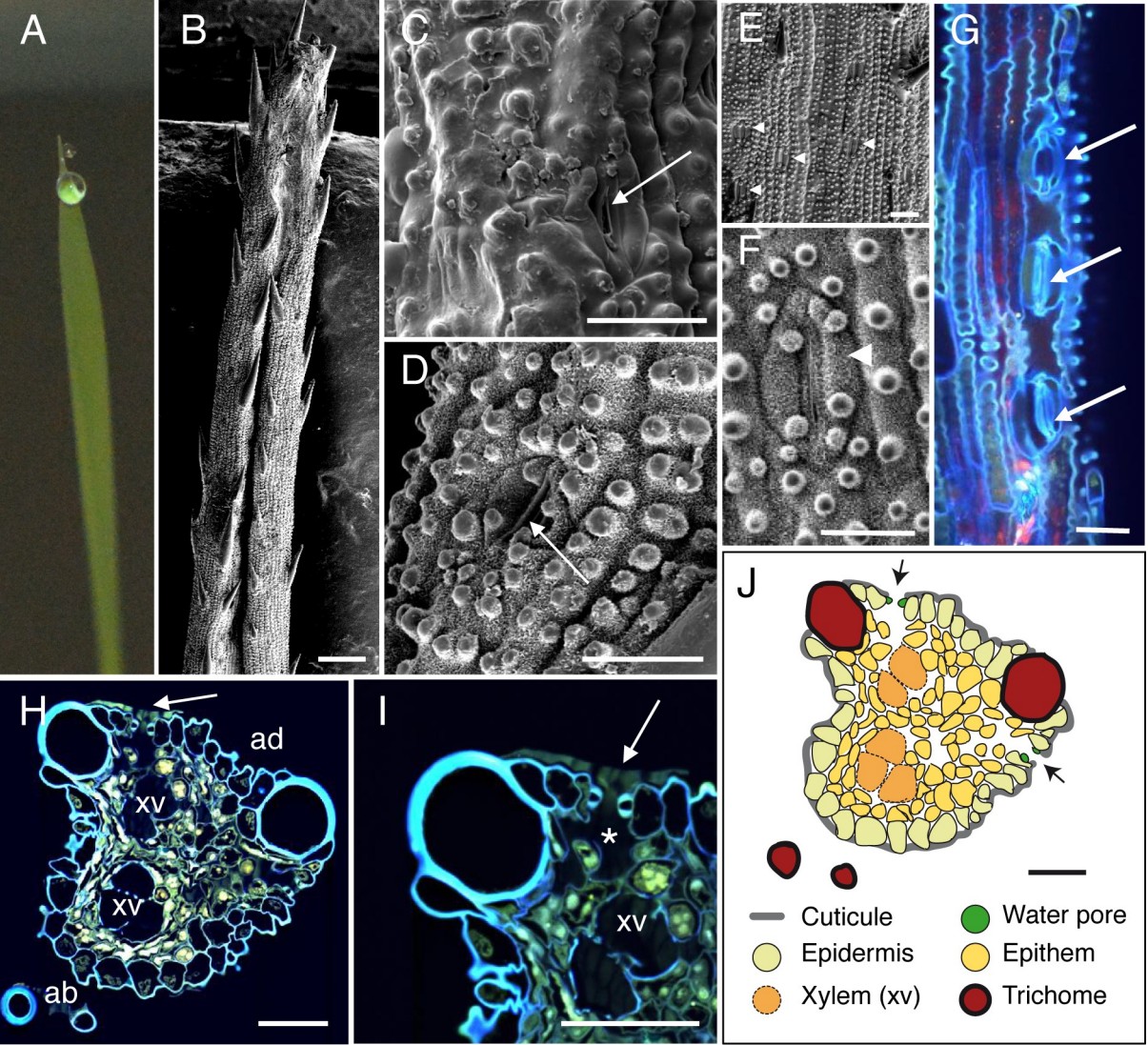

**Fig 2. Anatomic description of rice apical hydathodes by confocal and Scanning Electron Microscopy (SEM).** (A) A guttation droplet at the leaf tip. (B-D) The leaf tip was imaged by SEM. Water pores observed at the tip (C) and 300 μm from the extremity (D). (E-F) Observations of stomata were performed on distal parts of leaves relative to panels B-D. Panel F is a closeup image of panel E. (G) Confocal images of fresh tissue at 80–100 μm from the tip. The image is a maximal projection of 140 confocal planes in z dimension (1-μm steps). (H-I) Transversal sections (1-μm thickness) of fixed tissue at 50 μm from the tip were observed by confocal microscopy. White arrows, arrowheads and asterisk indicate water pores, stomata and large chambers and intercellular spaces, respectively. ad, adaxial face; ab, abaxial face; xv, xylem vessels. (J) Schematic drawing of the hydathode cross section observed in H. Scale bars: B: 100 μm; E: 20μm; D: 10 μm; C-D, G-J: 30 μm.

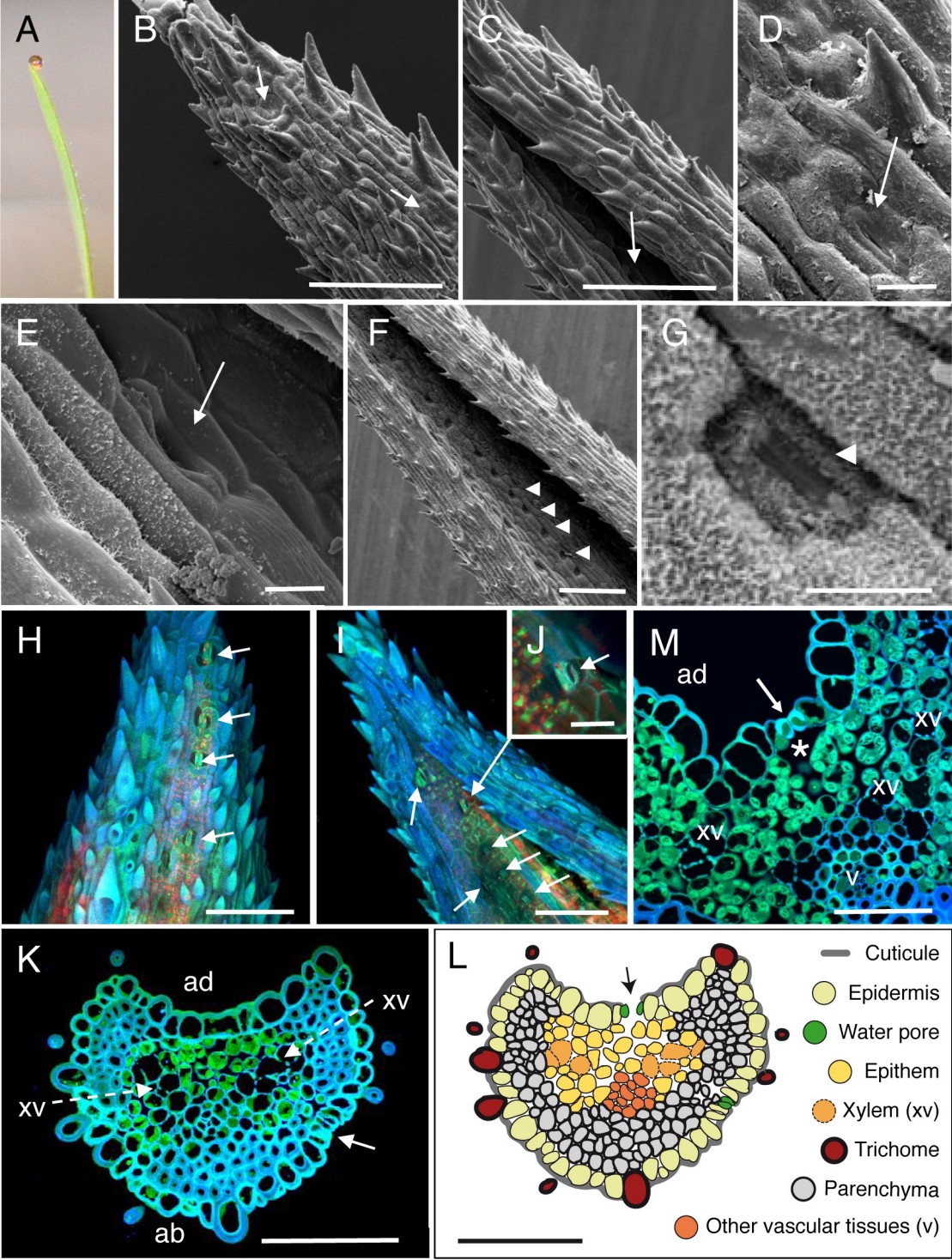

**Fig 3. Anatomic description of *Brachypodium distachyon* apical hydathodes by confocal and Scanning Electron Microscopy (SEM).** (A) A guttation droplet at the leaf tip. (B-E) The leaf tip was imaged by SEM. Water pores observed on the abaxial (B, D) and adaxial (C, E) faces of the leaf tip. (F-G) Observations of stomata were performed on distal parts of leaves relative to panels B-E. Panel G is a closeup image of panel E. (H-J) Confocal images of fresh leaf tips on their abaxial (H) and adaxial (I, J) faces. (K, M) Transversal sections (1-μm thickness) of fixed tissue at 60–70 μm (K) and 200 μm (M) from the tip were observed by confocal microscopy. White arrows, arrowheads, dashed arrows and asterisk indicate water pores, stomata, xylem vessels (xv) and large chambers and intercellular spaces, respectively. v, small veins; ad, adaxial face; ab, abaxial face. (L) Schematic drawing of an hydathode cross section as observed in K. Scale bars: B-C, F, H-I: 100 μm; K-M: 50 μm; J: 20 μm; D-E, G: 10 μm.

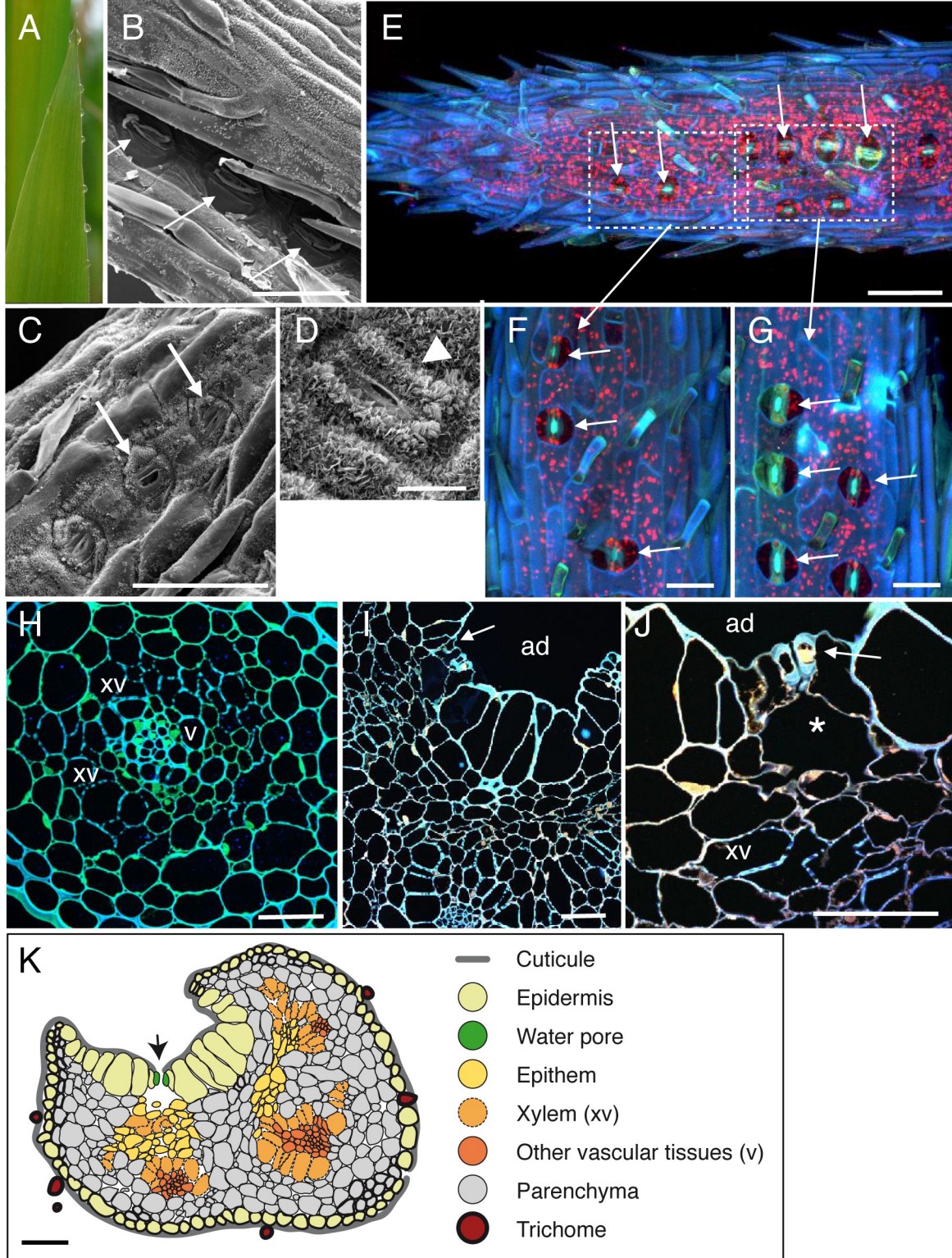

**Fig 4. Anatomic description of sugarcane apical hydathodes by confocal and Scanning Electron Microscopy (SEM).** (A) A guttation droplet at the leaf tip. (B-C) The leaf tip was imaged by SEM on the adaxial face at 200 μm (B) and 300 μm (C) from the spike tip. Water pores can be observed. (D) Observation of a stomate was performed on distal parts of leaves relative to panels B-C. (E-G) Confocal images of fresh leaf tip at 250–300 μm from the spike tip. Details from water pores (F-G). (H-I) Transversal sections (1-μm thickness) of fixed tissue at 150–200 μm (H) and 800 μm (I) from the spike tip were observed by confocal microscopy. White arrows, arrowheads and

asterisk indicate water pores, stomata and large chambers and intercellular spaces, respectively. ad, adaxial face; ab, abaxial face; xv, xylem vessels. (K) Schematic drawing of a hydathode cross section observed at 800 μm from the spike. Scale bars: E, K: 100 μm; B, C: 50 μm; I-J: 40 μm; F-H: 30 μm, D: 10 μm.

tip of the leaf where guttation happens; their irregular distribution on the leaf surface compared to stomata; their insertion below the epidermal layer surface, forming a depression compared to the neighbouring epidermal cells; their ticker guard cells compared to stomata; their opened mouth though some may be occasionally closed or obstructed; the lack or low accumulation of cuticular waxes at the pore and neighbouring cell wall surfaces compared to stomata. Those morphological features of water pores are all the more pronounced as the pores are close to the apex suggesting a common developmental origin of water pores and stomata and a later differentiation influenced by positional or environmental cues.

## Anatomy of monocot apical hydathodes is characteristic of epithemal hydathodes

The inner organization of hydathodes was first investigated by confocal microscopy on fresh samples using the autofluorescence of cell walls. In maize, chambers and xylem vessels could be visualized below the water pores (Fig 1G–1I). Yet, collecting information on the organization of the rest of the tissue remained challenging likely due to the limited cell wall fluorescence. Transversal thin sections from fixed samples were thus prepared and observed to refine the cytology of inner tissues. In the four monocot species studied, we confirmed the presence of a chamber below each water pore (Figs 1J–1L, 2H-2J, 3L, 3M and 4I-4K). The surrounding tissues are formed by loosely packed parenchyma cells with numerous intercellular spaces which form a continuous network from the water pore chamber to the vascular elements. In contrast to the leaf vasculature, hydathode vasculature is more disorganized and assembled into groups of two and more xylem vessels each which are surrounded neither by a bundle sheath nor by a layer of thickened-wall cells (Figs 1J–1L, 2H-2J, 3K-3M and 4H-4K). Such organization is typical of epithemal hydathodes as defined [1]: epidermal water pores, subwater pore chambers, loose small-celled parenchyma called epithem tightly and directly connected to an abundant vasculature.

Some levels of variation in this organization pattern can be observed (See Table 1 and schematic drawings in Figs 1K, 2J, 3L and 4K). In maize and rice, the epithem occupies with the vasculature the whole inner space of the hydathodes. In rice, the epithem is sometimes so reduced (Fig 2H–2J) so that the connection between the water pores and the vascular elements is sometimes direct (Fig 2I). In sugarcane and *Brachypodium*, the hydathode tissues are embedded in a parenchyma distinct from the epithem and made of larger and more compact cells with occasional (sugarcane) or systematic (*Brachypodium*) cell wall re-enforcements (Figs 3K–3M, 4H and 4K). In all instances, the plant vasculature remains easily connected to the outside thanks to the absence of a bundle sheath and the lose organization of the epithem thus allowing a free apoplastic flow of guttation fluid from the xylem vessels to the water pores.

## Loss of pit membrane integrity can be observed in some xylem vessels inside apical hydathodes

In order to better observe the cell wall of xylem vessels within hydathodes, we used transmission electron microscopy (TEM) coupled to PATAg labelling of cell wall polysaccharides (S1 Fig). Xylem vessel elements were identified by the presence of their lignified secondary cell wall thickenings. Cell wall thickenings were most developed in maize (S1A–S1B Fig) compared

Table 1. Main characteristics of tissues from leaf apical hydathodes of maize, rice, *Brachypodium distachyon* and sugarcane.

| Tissue | Maize var. P1524 | Rice var. Kitaake | *Brachypodium distachyon* | Sugarcane var HOCP04838, Q155 or CAS2 |
|---|---|---|---|---|
| Leaf tip shape | Gutter-like | Highly rolled | Rolled | Needle-like spike to gutter-like |
| Location of hydathodes | Apical and marginal | Apical | Apical | Sub-apical and marginal |
| Epidermis | No or few trichomes; epicuticular waxes | Many trichomes and spicules; epicuticular waxes | Many trichomes; epicuticular waxes | Some trichomes; epicuticular waxes |
| Location of water pores | At leaf tip and along leaf margin over several mm | At leaf tip and gutter | At leaf tip and gutter | Below the spike and along leaf margin over several cm |
| Epiculticular waxes on water pores | None or little | Little | Little | None or little |
| Other features of water pores | Often opened, absence of subsidiary cells | | | |
| Chambers below water pores | Large | Large | Small | Large |
| Epithem | Loose tissue; thin cell walls, numerous meatuses | Very reduced and loose | Reduced but compact; surrounded by a compact parenchyma with thick cell walls | Reduced but compact; surrounded by a compact parenchyma with sometime thickened cell walls |
| Connection of vessel elements to water pore chambers | Separated by few epithemal cells; connection via meatuses | Sometimes direct | Separated by some epithemal cells | Separated by some epithemal cells |

to rice, sugarcane or *Brachypodium* (S1C–S1J Fig). Between these ornamentations, a cell wall ca. ten times thinner than the primary cell wall of the neighbouring epithem parenchyma cells is observed and called pit membrane. PATAg labelling of these pit membranes is heterogeneous and discontinuous (S1D, S1H and S1J Fig) indicative of either a distinct polysaccharidic composition of these domains or the absence of any cell wall barriers. Altogether, these observations suggest that fluids meet limited physical barriers and likely flow freely across the cell wall of the xylem vessels.

## Discussion

### Variations on the theme of epithemal hydathodes in monocots

In monocots, guttation is always observed at leaf tips and sometimes at leaf margins such as in maize and sugarcane (Fig 4) [1]. Apical hydathodes were easy to identify at the leaf apex where the vasculature converges. Previous observations of monocots hydathodes by light microscopy and sometimes by scanning and transmission electron microscopy were often partial and limited to rice, barley and wheat [6, 17, 19–23]. Here, a full set of microscopic techniques was used yielding a comprehensive description of both surface and inner anatomy of hydathodes in rice and three additional monocot plants (Table 1). Our study confirmed some of the observations made in rice and revealed the conservation of several features of epithemal hydathodes in monocots: reduced wax apposition on the epidermis, opened water pores morphologically distinct from stomata, presence of a reduced epithem [17, 19] and dense xylem system. Main differences besides leaf curling were the shape of the leaf tip, the hydathode surface (presence of trichomes or spicules . . .), the size of hydathodes or the abundance of epithem cells relative to the parenchyma.

### A developmental gradient: From water pores to stomata

Mutations affecting stomatal development often similarly affect water pore development [24]. Several markers for stomatal identity or differentiation do not differentiate water pores from

stomata either [9], thus suggesting a common origin of both cell types. Yet, several morphological differences can distinguish water pores from stomata. Water pores are often inserted deeper in the epidermis. Also, the subsidiary cells known to be important for stomatal movement [25] could not be observed around water pores. Yet, the transition from water pores to stomata is not as dramatic in monocots as in dicots [9, 26] and it is thus sometimes difficult to locate the hydathode boundaries in monocots. We could observe a morphological gradient between water pores at the leaf apex and stomata in more distal areas of the leaf. Auxin is a good candidate for the establishment of this gradient since auxin maxima and expression of auxin biosynthetic genes is observed in rice hydathodes [27] similar to dicots [28, 29]. Because auxin was recently described as a negative regulator of stomatal differentiation [30], it remains to be experimentally tested whether auxin accumulating at hydathodes could impact water pore differentiation and be responsible for the observed developmental gradient.

## Morphological adaptations of monocot hydathodes driving guttation

Monocots hydathodes exhibit typical features of epithemal hydathodes. In such hydathodes, water transport is passive and driven by root pressure [for review, see 1]. Thus, guttation is likely favoured by morphological adaptations such as the absence of bundle sheath between the xylem vessels and the epithem, the thin cell walls of xylem vessels, the reduced epithem with many lacunas, the water pores and the cup shape of the leaf. These features are not specific of monocots and some can be found in dicots [1]. Similar to cauliflower and Arabidopsis [9], reduced epicuticular wax depositions are observed at hydathodes compared to the leaf blade which could help preventing guttation droplets from falling. These surface properties come in addition to leaf shape adaptations such as grooves, trichomes or indentations which should also favour droplet formation and accumulation at hydathodes.

## Monocot hydathodes can provide facilitated access to plant vasculature for microbial pathogens

Leaf surface properties such as the cuticle and epicuticular waxes also strongly affect microbial adhesion, behaviour and survival in the phyllosphere [for review, see 31]. For instance, leaf wettability in maize is positively correlated to the charge in epiphytic bacteria [32]. Thus, rain or spray irrigation may concentrate microbes at hydathodes. While hydathode anatomy likely offers little resistance to water fluxes, it also represents a potential breach which could be exploited by pathogens to access plant inner tissues, including the vasculature. For instance, we describe that rice xylem vessels are almost directly accessible once through water pores. Besides, we also observe holes in the pit membrane of xylem vessels in *Brachypodium* or sugarcane giving a facilitated access to the vasculature. Thus, xylem vessels within monocot hydathodes seem a lot more vulnerable to infection compared to dicot hydathodes where the epithem tissue is much more developed. The number of pathogens able to infect monocot hydathodes is also likely underestimated since *X. albilineans*, the causal agent of leaf scald in sugarcane [33, 34] and *X. translucens*, the causal agent of bacterial leaf streak on a broad host range of cereal crops and grasses [35–38] both cause symptoms starting from leaf tips or leaf margins.

To conclude, we described apical hydathodes from four monocot species. Besides species-to-species and leaf-to-leaf variations, we recognized anatomical features typical of epithemal hydathodes. The presence of grooves and trichomes and lower wax apposition at apical hydathodes seem adapted to hold guttation droplets at leaf tips. Open water pores provide an almost direct access the vascular elements due to a sometime reduced epithem and a thin

primary cell wall of xylem vessels. Our analyses form the basis for further investigations on the physiology and the immunity of hydathodes in those monocot plants.

## Materials and methods

### Plant material and growth conditions

The following plant species were studied: *Brachypodium distachyon*, rice (*Oryza sativa* var. Kitaake), sugarcane (*Saccharum officinarum x Saccharum spontaneum* hybrid, *var* HOCP04838, Q155 or CAS2) and maize (*Zea mays* var. P1524, Pioneer Dupont). The position of the leaf used for microscopy for *Brachypodium distachyon* and sugarcane was not possible to determine. For maize and rice, hydathodes of the second leaf were observed.

### Scanning Electron Microscopy (SEM)

Leaf samples were fixed under vacuum for 30 min with 2.5% glutaraldehyde in 0.2 M sodium cacodylate buffer (pH 7.2) containing 0.1% triton X-100 and at atmospheric pressure for 1h in the same solution without Triton X-100. Samples were dehydrated in a series of aqueous solutions of increasing ethanol concentrations (25, 50, 70, 95, 100%, 1 h each) and then critical-point dried with liquid $CO_2$. Samples were attached with double-sided tape to metal stubs grounded with conductive silver paint and sputter-coated with platinum. Images were acquired with a scanning electron microscope (Quanta 250 FEG FEI) at 5kV with a working distance of 1 cm.

### Optical and transmission electron microscopy

Preparation of hydathode samples for both optical and transmission electron microscopy were previously detailed [9, 39]. To observe fresh samples, leaf tips (1.5 cm in length) were mounted in water on a glass slide and covered with a coverslip. Images were acquired with a laser scanning confocal microscope (LSCM, Leica SP2 AOBS, Mannheim, Germany). To perform hydathode sections, leaf tips were fixed under vacuum for 30 min with 2.5% glutaraldehyde in 0.2 M sodium cacodylate buffer (pH 7.2) containing 0.1% triton X-100 and then at the atmospheric pressure for 1h in the same solution without triton X-100. The samples were then rinsed in the same cacodylate buffer, dehydrated in a series of aqueous solutions of increasing ethanol concentrations and infiltrated step-wise in LR White resin. They were finally polymerized for 24h at 60˚C. From embedded material, thin (1 μm in thickness) or ultra-thin (80–90 nm in thickness) sections were prepared using an UltraCut E ultramicrotome equipped with a diamond knife (Reichert-Leica, Germany). Transversal thin sections were used to acquire images with LSCM. All confocal images are the overlay of blue (410–470 nm), green (500–580 nm) and red (650–750 nm) channels used to depict the autofluorescence of the cell walls (blue and green channels) and of the chlorophyll (red channel) after excitation using a 405-nm diode laser. For transmission electron microscopy (TEM), ultra-thin sections were collected on gold grids and submitted to the periodic acid-thiocarbohydrazide-silver proteinate reaction (PATAg). PATAg staining of polysaccharides was used to enhance contrast and observe xylem ornamentations and pit membranes. Images were acquired using a Hitachi-HT-7700 (Japan) transmission electron microscope operating at 80 kV.

## Supporting information

**S1 Fig.** Observation of pit membranes integrity in hydathodes of maize (A-B), rice (C-D), Brachypodium (E-F) and sugarcane (G) by transmission electron microscopy.
(PDF)

## Acknowledgments

We are grateful to Jean-Heinrich Daugrois, Monique Royer (CIRAD, Montpellier, France) and Pam Ronald (UC Davis, CA) for contributing fixed sugarcane material and rice seeds, respectively. We wish to thank Jean-Marc Routaboul (INRA Toulouse) for critically reading the manuscript. Electron microscopy was performed on the CMEAB (Centre de Microscopie Electronique Appliquée à la Biologie) core facility with the expert assistance of Isabelle Fourquaux et Bruno Payre (Université Toulouse III Paul Sabatier, Toulouse, France).

## Author Contributions

**Conceptualization:** Alain Jauneau, Aude Cerutti, Laurent D. Noël.

**Data curation:** Alain Jauneau, Aude Cerutti, Laurent D. Noël.

**Formal analysis:** Alain Jauneau, Aude Cerutti.

**Funding acquisition:** Laurent D. Noël.

**Investigation:** Alain Jauneau, Aude Cerutti, Marie-Christine Auriac.

**Methodology:** Alain Jauneau, Marie-Christine Auriac.

**Project administration:** Laurent D. Noël.

**Supervision:** Alain Jauneau, Laurent D. Noël.

**Visualization:** Laurent D. Noël.

**Writing – original draft:** Alain Jauneau, Aude Cerutti, Laurent D. Noël.

**Writing – review & editing:** Alain Jauneau, Aude Cerutti, Marie-Christine Auriac, Laurent D. Noël.

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
