## [Decision Letter · Decision Letter 0]

14 Jul 2020

PONE-D-20-11070

Anatomy of epithemal hydathodes in four monocotyledon plants of economic and academic relevance

PLOS ONE

Dear Dr. NOEL,

Thank you for submitting your manuscript to PLOS ONE. After careful consideration, we feel that it has merit but does not fully meet PLOS ONE’s publication criteria as it currently stands. Therefore, we invite you to submit a revised version of the manuscript that addresses the points raised during the review process.

We look forward to receiving your revised manuscript.

Kind regards,

Mehdi Rahimi, Ph.D.

Academic Editor

PLOS ONE

Journal Requirements:

Reviewers' comments:

Reviewer's Responses to Questions

**Comments to the Author**

1. Is the manuscript technically sound, and do the data support the conclusions?

Reviewer #1: Yes

Reviewer #2: Yes

2. Has the statistical analysis been performed appropriately and rigorously? 

Reviewer #1: N/A

Reviewer #2: N/A

3. Have the authors made all data underlying the findings in their manuscript fully available?

Reviewer #1: Yes

Reviewer #2: Yes

4. Is the manuscript presented in an intelligible fashion and written in standard English?

Reviewer #1: Yes

Reviewer #2: Yes

5. Review Comments to the Author

Reviewer #1: The authors present a detailed description of hydathodes in four monocots (maize, rice, sugarcane and Brachypodium distachyon) using confocal microscopy as well as scanning and transmission electron microscopy. First, the authors compare the location of the guttation droplets and the water pores on the surface of the leaves. Furthermore, they describe the inner organization of the hydathodes for these four species. It is to my knowledge the first study in which such efforts have been put into the observation and the comparison of hydathodes anatomy in different species.

Major points for revision:

-The manuscript will benefit from a comparative Table that recapitulates the whole of observations made for the four different species.

- The paper will benefit from putting Fig S1 directly into Fig 1-4 and compare for the SEM images side-by-side for stomata and hydathode. One can add the observed differences as text below the two SEM images to make it more clear to the reader (maize with maize etc).

-It is sometimes difficult to follow what the authors claim from the microscopical pictures from what we actually see, especially on the pictures from confocal microscopy on fresh leaves. For example, in the figure 1 F and G, the authors claim the presence of a chamber and xylem vessels under maize water pores. Are F and G pictures from the same hydathode at different depth? Although the presence of xylem is obvious, the presence of a chamber is not, thus authors should rather cite their figure 1 H or J.

-It would be helpful to have more homogeneous legend between the microscopical pictures and the schematic drawing. For instance, by adding the xv, v, *, … short legend names to the schematic drawings. Additionally, you might want to add xv to the dotted arrow on Figure 1H

-The authors enumerated a list of criteria differentiating stomata from water pores (from line 99 on). Where are these criteria coming from? Are they criteria obtained from their observations? It is difficult to evaluate several of these criteria from the presented figures (accumulation of cuticular waxes, for instance). How did the authors differentiated the water pores from the stomata in confocal microscopy on fresh leaves? Because none of the criteria listed seems to be obvious from the presented confocal pictures. I think that it is visible in the SEM images, but it will be handier if we can directly compare the images side-by-side with text that defines the differences.

Minor points for revision:

-Title, Is there non epithemal hydathodes? Should it be changed for (apical) Hydathodes? It is mentioned epithemal hydathodes several times in the manuscript but not defined anywhere. For example, in the result subtitle (line 150): “Anatomy of monocot apical hydathodes is characteristic of epithemal hydathodes”. Please explain in the introduction the difference between hydathodes and epithemal hydathode.

-L38 humidity [is] high; add ‘is’

-L44 Aren’t hydathodes all on the leaf surface? Change for ‘leaf blade’.

-L53 and L58: To my knowledge the hydathode is targeted primarily by bacterial pathogens that can colonize the vasculature. Please add bacterial. Virus, oomycetes and fungi appear to have different entry routes.

-L55 Name the causal agent(s) of Brassica black rot as you do for the other diseases.

-L80 SEM observations (change in plural)

-L85 delete: ‘the frame of’

-L96 figure 4 D rather than figure 3 B and C

-L104-106: Sentence is unclear: “Those changes are gradual along the leaf longitudinal axis suggesting a common developmental origin of both cell types influenced by positional or environmental cues.“

Which changes [differences…]? What do the authors means by gradual? Is it more pronounced at the tip, less pronounced? Please clarify.

- L185-186: cryptic sentence (limiting –> that limit), barriers -> physical barriers?

- L191 at [the] leaf apex

- L220 Monocots : remove S

- L212 “developmental gradient” is a vague term. I presume that the authors mean a gradient across the leaf from tip-to-base? Morphological traits [signature…]? Trait is more a genetic term to my knowledge. I would again argue that a table that summarises the differences can be helpful. This table can also contain these cell type specific signatures.

L213-214 cryptic sentence. Please rephrase. I guess that they mean that evidence exists that auxin maxima appear to occur in hydathodes meaning that this might be an organizing principle for hydathode development compared to stomatal development?

- L220-223 What is the evidence that these features support guttation? There is currently no reference. It is unknown if guttation is an active process that is facilitated by the epithem cells (active water transport, like in the Casparian strip) or a passive flow of xylem sap from the xylem when root pressure exceeds water respiration.

- L280-284 I miss information on the dyes used to stain the cell wall for CSLM. Did they use any dyes to stain the cellulose etc (Calcofluor-white)?

Reviewer #2: The manuscript reports on microscopic examination of monocot hydathodes of four crops namely rice, maize, sugarcane and Brachypodium highlighting their similarities and specificities. It is a good piece of work and written well. However, the authors should undertake minor modifications as follows:

1. Provide legend/caption for all four figures;

2. The scale for microscopic examination should also be shown in the figure itself or at least in the figure caption.

3. Only two references from last two years. Authors may think of providing few more relevant recent references from last two years.

6. PLOS authors have the option to publish the peer review history of their article (what does this mean?). If published, this will include your full peer review and any attached files.

Reviewer #1: No

Reviewer #2: **Yes: **Dr Charu Lata

---

## [Author Response · Author response to Decision Letter 0]

29 Jul 2020

Reviewer #1: The authors present a detailed description of hydathodes in four monocots (maize, rice, sugarcane and Brachypodium distachyon) using confocal microscopy as well as scanning and transmission electron microscopy. First, the authors compare the location of the guttation droplets and the water pores on the surface of the leaves. Furthermore, they describe the inner organization of the hydathodes for these four species. It is to my knowledge the first study in which such efforts have been put into the observation and the comparison of hydathodes anatomy in different species.

Major points for revision:

-The manuscript will benefit from a comparative Table that recapitulates the whole of observations made for the four different species.

#We agree. A new Table 1 now reports on the shared and distinct properties of hydathode tissues in the four species studied.

- The paper will benefit from putting Fig S1 directly into Fig 1-4 and compare for the SEM images side-by-side for stomata and hydathode. One can add the observed differences as text below the two SEM images to make it clearer to the reader (maize with maize etc).

#As suggested, Figure S1 has been incorporated in Fig 1-4 to facilitate the comparisons.

-It is sometimes difficult to follow what the authors claim from the microscopical pictures from what we actually see, especially on the pictures from confocal microscopy on fresh leaves. For example, in the figure 1 F and G, the authors claim the presence of a chamber and xylem vessels under maize water pores. Are F and G pictures from the same hydathode at different depth? Although the presence of xylem is obvious, the presence of a chamber is not, thus authors should rather cite their figure 1 H or J.

#We carefully re-examined these points and modified the text whenever needed. Fig.1F and G are indeed the observations of the same sample along the z axis as already mentioned in the figure caption. We agree that Fig 1H is better suited to observe chambers below the pores while panel F and G illustrate the close vicinity of water pores and xylem vessels.

-It would be helpful to have more homogeneous legend between the microscopical pictures and the schematic drawing. For instance, by adding the xv, v, *, … short legend names to the schematic drawings. Additionally, you might want to add xv to the dotted arrow on Figure 1H

#We have now included the abbreviations in the drawings to facilitate interpretations and comparisons with the corresponding cross-sections.

-The authors enumerated a list of criteria differentiating stomata from water pores (from line 99 on). Where are these criteria coming from? Are they criteria obtained from their observations? It is difficult to evaluate several of these criteria from the presented figures (accumulation of cuticular waxes, for instance). How did the authors differenciated the water pores from the stomata in confocal microscopy on fresh leaves? Because none of the criteria listed seems to be obvious from the presented confocal pictures. I think that it is visible in the SEM images, but it will be handier if we can directly compare the images side-by-side with text that defines the differences.

#These criteria differentiating water pores from stomata are arising from our observations. This has now been rephrased. Our observations are reminiscent of observations made in other plants (see discussion). Confocal images were particularly useful for the imaging of the aperture of pores compared to stomata in living tissues. Comparing LCSM and SEM images was important to rule out possible artefacts on fixed tissues. As suggested by the reviewer, including Fig S1 into Fig 1-4 will help the reader do visualize the differences.

Minor points for revision:

-Title, Is there non epithemal hydathodes? Should it be changed for (apical) Hydathodes? It is mentioned epithemal hydathodes several times in the manuscript but not defined anywhere. For example, in the result subtitle (line 150): “Anatomy of monocot apical hydathodes is characteristic of epithemal hydathodes”. Please explain in the introduction the difference between hydathodes and epithemal hydathode.

#We agree that “leaf apical” hydathode would be more appropriate for a broad audience in the title: we made the change. Historically, hydathodes were classified into passive and active hydathodes. In passive hydathodes, secretion is driven by the root pressure. Anatomy comprises epidermal water pores and an epithem irrigated by the xylem. In contrast, active hydathodes are related to glands and trichomes. In nature, a continuum between both types of hydathodes can be observed. We now define both hydathode types in the introduction (Lines 36-40)

-L38 humidity [is] high; add ‘is’

#Corrected

-L44 Aren’t hydathodes all on the leaf surface? Change for ‘leaf blade’.

#Corrected

-L53 and L58: To my knowledge the hydathode is targeted primarily by bacterial pathogens that can colonize the vasculature. Please add bacterial. Virus, oomycetes and fungi appear to have different entry routes.

#Corrected

-L55 Name the causal agent(s) of Brassica black rot as you do for the other diseases.

#Corrected

-L80 SEM observations (change in plural)

#Corrected

-L85 delete: ‘the frame of’

#Corrected

-L96 figure 4 D rather than figure 3 B and C

#Corrected

-L104-106: Sentence is unclear: “Those changes are gradual along the leaf longitudinal axis suggesting a common developmental origin of both cell types influenced by positional or environmental cues.“

#Which changes [differences…]? What do the authors means by gradual? Is it more pronounced at the tip, less pronounced? Please clarify.

We meant that water pore anatomical features are all the more pronounced as the pore is close to the apex. The sentence was rephrased to be more explicit

- L185-186: cryptic sentence (limiting –> that limit), barriers -> physical barriers?

#The sentence was rephrased to be more explicit

- L191 at [the] leaf apex

#Corrected

- L220 Monocots : remove S

#Corrected

- L212 “developmental gradient” is a vague term. I presume that the authors mean a gradient across the leaf from tip-to-base? Morphological traits [signature…]? Trait is more a genetic term to my knowledge. I would again argue that a table that summarises the differences can be helpful. This table can also contain these cell type specific signatures.

#We refer here to a morphological continuum between water pores and stomata. “Traits” has been removed. Table 1 has been introduced to summarize our main observations.

- L213-214 cryptic sentence. Please rephrase. I guess that they mean that evidence exists that auxin maxima appear to occur in hydathodes meaning that this might be an organizing principle for hydathode development compared to stomatal development?

#Agreed. The sentence was rephrased to be more explicit.

- L220-223 What is the evidence that these features support guttation? There is currently no reference. It is unknown if guttation is an active process that is facilitated by the epithem cells (active water transport, like in the Casparian strip) or a passive flow of xylem sap from the xylem when root pressure exceeds water respiration.

#Monocot hydathodes described here are epithemal (also called passive) hydathodes. Water transport is passive, driven by root pressure and is not dependent of active water secretion by the epithem tissue of the hydathode as known in active hydathodes. As mentioned by the reviewer, the presence of the casparian strip and bundle sheath is key to drive fluids towards passive hydathodes, resulting in guttation. For instance, an Arabidopsis vascular bundle sheath mutant fails to guttate (Shatil-Cohen A, Moshelion M. 2012. Smart pipes: the bundle sheath role as xylem-mesophyll barrier. Plant Signal. Behav. 7:1088–91). Obstructing hydathodes also results in mesophyll flooding in Chloranthus japonicus (Field TS, Sage TL, Czerniak C, Iles WJD. 2005. Hydathodal leaf teeth of Chloranthus japonicus(Chloranthaceae) prevent guttation-induced flooding of the mesophyll. Plant Cell Environ. 28:1179–90). The sentence was rephrased to be more explicit.

- L280-284 I miss information on the dyes used to stain the cell wall for CSLM. Did they use any dyes to stain the cellulose etc (Calcofluor-white)?

#Images from CSLM did not use any dye and correspond to the imaging of the autofluorescence of cell walls (blue and green channels) and chlorophyll (red channel) after excitation at 405 nm. The phrasing has been modified to be more explicit.

Reviewer #2: The manuscript reports on microscopic examination of monocot hydathodes of four crops namely rice, maize, sugarcane and Brachypodium highlighting their similarities and specificities. It is a good piece of work and written well. However, the authors should undertake minor modifications as follows:

1. Provide legend/caption for all four figures;

#Following PLoS one manuscript formatting instructions, Figure captions were included in the main text body after the result section in which they were first cited, i.e. lines 108-148 in the first version of the manuscript.

2. The scale for microscopic examination should also be shown in the figure itself or at least in the figure caption.

#Scales were indicated at the end of every figure caption.

3. Only two references from last two years. Authors may think of providing few more relevant recent references from last two years.

#Hydathode literature is very scarce and mostly dating from the 70s or earlier. We believe that all recent relevant publications were already cited.

---

## [Decision Letter · Decision Letter 1]

1 Sep 2020

Anatomy of leaf apical hydathodes in four monocotyledon plants of economic and academic relevance

PONE-D-20-11070R1

Dear Dr. NOEL,

We’re pleased to inform you that your manuscript has been judged scientifically suitable for publication and will be formally accepted for publication once it meets all outstanding technical requirements.

Kind regards,

Mehdi Rahimi, Ph.D.

Academic Editor

PLOS ONE

Additional Editor Comments (optional):

Reviewers' comments:

Reviewer's Responses to Questions

**Comments to the Author**

1. If the authors have adequately addressed your comments raised in a previous round of review and you feel that this manuscript is now acceptable for publication, you may indicate that here to bypass the “Comments to the Author” section, enter your conflict of interest statement in the “Confidential to Editor” section, and submit your "Accept" recommendation.

Reviewer #1: All comments have been addressed

Reviewer #2: All comments have been addressed

2. Is the manuscript technically sound, and do the data support the conclusions?

Reviewer #1: Yes

Reviewer #2: Yes

3. Has the statistical analysis been performed appropriately and rigorously? 

Reviewer #1: N/A

Reviewer #2: N/A

4. Have the authors made all data underlying the findings in their manuscript fully available?

Reviewer #1: Yes

Reviewer #2: Yes

5. Is the manuscript presented in an intelligible fashion and written in standard English?

Reviewer #1: Yes

Reviewer #2: Yes

6. Review Comments to the Author

Reviewer #1: All my suggestions have been addressed in a very satisfactory manner. This manuscript is a careful and detailed study on the anatomy of hydathodes in monocots using advanced and state-of-the-art microscopy tools.

Reviewer #2: (No Response)

7. PLOS authors have the option to publish the peer review history of their article (what does this mean?). If published, this will include your full peer review and any attached files.

Reviewer #1: No

Reviewer #2: No

---

## [Editor Report · Acceptance letter]

8 Sep 2020

PONE-D-20-11070R1 

Anatomy of leaf apical hydathodes in four monocotyledon plants of economic and academic relevance 

Dear Dr. NOEL:

I'm pleased to inform you that your manuscript has been deemed suitable for publication in PLOS ONE. Congratulations! Your manuscript is now with our production department. 

Kind regards, 

on behalf of

Dr. Mehdi Rahimi 

Academic Editor

PLOS ONE